# Bacteriophage specificity is impacted by interactions between bacteria

Ave T. Bisesi,[1] Wolfram Möbius,[2,3] Carey D. Nadell,[4] Eleanore G. Hansen,[5] Steven D. Bowden,[5] William R. Harcombe[1,6]

**ABSTRACT**  Predators play a central role in shaping community structure, function, and stability. The degree to which bacteriophage predators (viruses that infect bacteria) evolve to be specialists with a single bacterial prey species versus generalists able to consume multiple types of prey has implications for their effect on microbial communities. The presence and abundance of multiple bacterial prey types can alter selection for phage generalists, but less is known about how interactions between prey shape predator specificity in microbial systems. Using a phenomenological mathematical model of phage and bacterial populations, we find that the dominant phage strategy depends on prey ecology. Given a fitness cost for generalism, generalist predators maintain an advantage when prey species compete, while specialists dominate when prey are obligately engaged in cross-feeding interactions. We test these predictions in a synthetic microbial community with interacting strains of *Escherichia coli* and *Salmonella enterica* by competing a generalist T5-like phage able to infect both prey against P22*vir*, an *S. enterica*-specific phage. Our experimental data conform to our modeling expectations when prey species are competing or obligately mutualistic, although our results suggest that the *in vitro* cost of generalism is caused by a combination of biological mechanisms not anticipated in our model. Our work demonstrates that interactions between bacteria play a role in shaping ecological selection on predator specificity in obligately lytic bacteriophages and emphasizes the diversity of ways in which fitness trade-offs can manifest.

**IMPORTANCE**  There is significant natural diversity in how many different types of bacteria a bacteriophage can infect, but the mechanisms driving this diversity are unclear. This study uses a combination of mathematical modeling and an *in vitro* system consisting of *Escherichia coli*, *Salmonella enterica*, a T5-like generalist phage, and the specialist phage P22*vir* to highlight the connection between bacteriophage specificity and interactions between their potential microbial prey. Mathematical modeling suggests that competing bacteria tend to favor generalist bacteriophage, while bacteria that benefit each other tend to favor specialist bacteriophage. Experimental results support this general finding. The experiments also show that the optimal phage strategy is impacted by phage degradation and bacterial physiology. These findings enhance our understanding of how complex microbial communities shape selection on bacteriophage specificity, which may improve our ability to use phage to manage antibiotic-resistant microbial infections.

**KEYWORDS**  bacteriophages, virus-host interactions, microbial communities, microbial ecology, competition, mutualism

Address correspondence to William R. Harcombe, harcombe@umn.edu.

The authors declare no conflict of interest.

See the funding table on p. 19.

Predators can impose top-down control of ecosystems, impacting species abundances, community structure, and community function (1). For example, in marine environments, lytic bacteriophages (phages), the viral predators of bacteria, are critical

drivers of microbial populations and nutrient cycling, lysing up to 40% of phytoplankton biomass per day (2). The diet breadth of predators—how many different prey species they can consume—is an important component of how top-down control shapes an environment (Fig. 1A) (3–8). Specialist predators often drive limit cycles with their prey, while generalists are more likely to stabilize prey populations through the emergence of apparent mutualisms, in which the presence of one prey species reduces the burden of predation on the other (9–14). Diversity in specificity is widespread in microbial communities, where some phages are generalists that can prey upon bacterial species across multiple genera, while others specialize on a single serovar (15, 16). There are likely many determinants of phage specificity, particularly in coevolving communities, as bacterial prey develop mechanisms either to block phage adsorption (e.g., loss of receptors) or prevent phage replication (e.g., clustered regularly interspaced short palindromic repeats [CRISPR], superinfection immunity) (17) and phage evolve to overcome these resistance mechanisms, often with pleiotropic costs for host range expansion (18). Identifying the key forces shaping predator diet breadth, as well as their relative importance, would therefore have substantial consequences for our ability to predict the long-term dynamics of multitrophic microbial communities.

The composition of the prey community is one force known to impact predator specificity. The evolution of generalist predators often requires prey heterogeneity to provide opportunities for diversification (19–24). While it has been suggested that prey

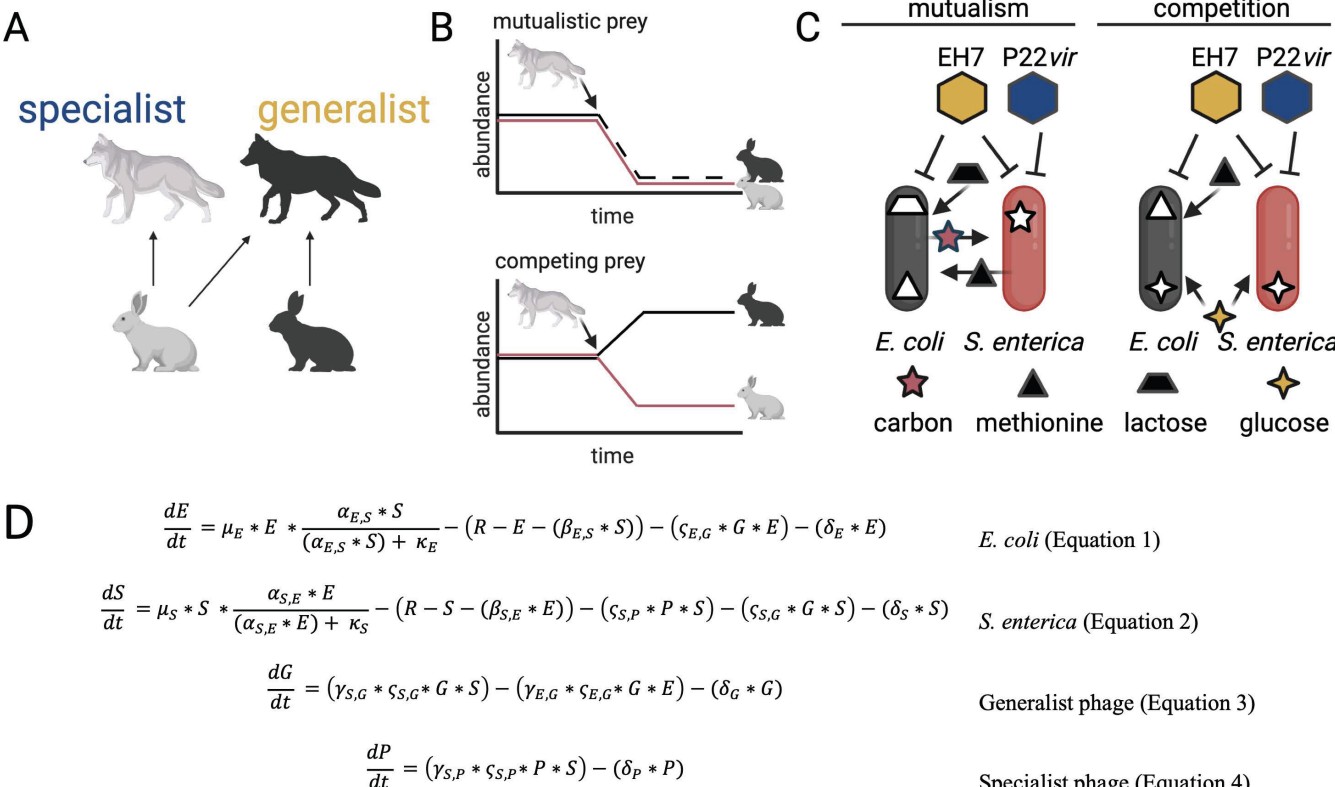

**FIG 1** Specificity in a microbial synthetic community. (A) Schematic of diet breadth in macropredators. Species with generalist diets have wide diet breadths spanning multiple resources, while specialists have narrower diet breadths, sometimes specific to a single resource. (B) Expected community dynamics when mutualistic or competing prey species are challenged by a specialist predator. When prey are mutualistic, predation will reduce abundances of both species. When prey compete, predation will reduce the abundance of one species and result in an increase in the abundance of the other species through competitive release. (C) Schematic of cross-feeding system consisting of an *Escherichia coli* methionine auxotroph and *Salmonella enterica* methionine secreter. In lactose minimal media, *E. coli* provides carbon byproducts to *S. enterica* and *S. enterica* provides methionine to *E. coli*. In glucose minimal media with methionine, the two bacteria compete. The phage P22*vir* is a specialist on *S. enterica*, while the phage EH7 is a generalist that can attack both bacterial species. (D) Lotka-Volterra style modeling equations. For additional details, see Materials and Methods. For panels A, B, and C: figure created with BioRender.

diversity could reduce the incidence of predator generalism given the demands of engaging in coevolutionary arms races with multiple species (25), microbial studies have shown that the presence of multiple bacterial strains is sufficient to select for generalists (19, 26). Assuming a heterogenous prey environment, optimal foraging theory provides several additional predictions of how the structure of prey communities might shape predator diet breadth. First, it suggests that absolute prey densities alter selection on predator specificity by impacting foraging time (27). Generalism is predicted to be favored at low prey densities when foraging time is high, while specialism is favored at high prey densities; this prediction has been validated in a microbial system (27). Optimal foraging theory also emphasizes the importance of relative prey abundances, such that predators should experience selection to exploit the most abundant prey types, even if those prey are low quality or intraspecific competition between predators is strong (22, 23, 28, 29). However, even when relative abundances are considered, most studies on diet breadth assume a static ratio of prey types over time, omitting a critical dimension of natural communities.

Interactions between prey complicate the assumption of static ratios by generating correlations between prey abundances (30–32). Nutrient competition between bacteria tends to generate anti-correlated abundances between species, while positive interactions such as obligate mutualism generate positively correlated abundances (29–32). When predators are consuming prey species with anti-correlated abundances, a generalist strategy is likely to be favored, because predation on one species should lead to an increase in the abundance of the alternative prey through competitive release (Fig. 1B). The expectation that competing prey should favor predator generalism is consistent with $R^*$ theory, which suggests that the outcome of competition on a shared resource is generally determined by which species can survive on the lowest levels of the resource (33). Given that a generalist predator has alternative resources available to it, it should always be able to survive on the lower levels of a resource shared with a specialist. Additional theoretical work provides support for the notion that there are situations in which competing prey should favor generalist predators (29, 34). When prey compete, mathematical modeling suggests that competitive dominance by a novel prey type is generally required for the evolution of broadened predator diet breadth when fitness trade-offs for generalism are present (29). Comparatively little work has been done investigating the impact of prey engaged in direct positive interactions on predator diet breadth. Positively correlated prey abundances are likely to favor specialist predators because predation by a specialist would also lead to a reduction in the abundance of the alternative prey (Fig. 1B). The interdependence between prey species may also increase the likelihood of overexploitation by predators (35–37). However, the hypothesis that different types of interactions between prey should alter predator specificity has not been fully validated theoretically or empirically.

Here, we use a mathematical model and an *in vitro* system to investigate how interactions between prey species govern ecological selection on predator diet breadth. We found that, in our model, obligately mutualistic interactions between microbial prey were more likely to favor a specialist phage predator, while competition between prey was more likely to favor a generalist phage predator. These findings were in accordance with our initial hypotheses and are relevant to systems where interactions between prey drive correlations in their abundance. We tested these findings in an experimental system (Fig. 1C) and reproduced our ecological modeling results despite differences in the mechanism of fitness trade-off experienced by the generalist phage. Our work provides insight into how interactions between two microbial prey species alter ecological selection on phage specificity in well-mixed environments and provides a foundation for predicting the evolution and maintenance of specificity in bacteriophage, with implications for designing and managing microbial communities.

## RESULTS

### In a phenomenological model, phage relative abundance depends on prey interactions and fitness trade-offs for phage generalism

We used a phenomenological model (Fig. 1D; Materials and Methods) to predict how communities of two interacting prey species respond to attack by predatory lytic phage during chemostatic growth. We predicted that competition between prey was likely to favor predator generalism by increasing temporal heterogeneity in resource availability (38, 39), while obligate mutualism between prey species would result in less temporal heterogeneity, as bacterial species would either occur together or not at all, likely favoring specialization (38).

We first investigated the behavior of the model with a single parameter set. Using our default parameters (Table 1), we examined cases in which phage were not present, or when only one phage type was present. When phage were not modeled in our system, prey species reached a 50:50 ratio at equilibrium (Fig. 2A, left panel). The introduction of a specialist resulted in competitive release of *Escherichia coli* when prey competed and correlated reductions in bacterial abundances when prey were mutualistic (Fig. 2A, middle panel). This behavior was consistent with our conceptual model (Fig. 1B). In comparison, the introduction of a generalist phage predator reduced the density of both prey species at equilibrium, keeping their abundances at a 50:50 ratio (Fig. 2A, right panel). We also considered the behavior of the model when phage phenotypes competed against one another. When phage were parametrically identical, the generalist was more prevalent regardless of prey interaction type (Fig. 2B and C, top panels), though interaction type altered the degree of dominance. This finding suggested the importance of the "jack of all trades, master of none" hypothesis for the predominance of specialization, which asserts that, all other things equal, a cost of generalism is required to favor a specialist (40–46). Following the incorporation of a cost of generalism by increasing the specialist's burst size to five times greater than that of the generalist, we found that our modeling results were consistent with our conceptual model such that the generalist dominated when prey competed (Fig. 2B), while the specialist dominated when prey were mutualistic (Fig. 2C).

**TABLE 1** Dimensionless phenomenological model parameters, default values, and descriptions

| Parameter | Competition default value | Mutualism default value | Description |
|---|---|---|---|
| $\alpha_{E,S}$ | 1 | 1 | Mutualistic coefficient, benefit of prey species S to prey species E |
| $\beta_{E,S}$ | 1 | 0 | Competition coefficient, effect of prey species S on prey species E |
| $\alpha_{S,E}$ | 1 | 1 | Mutualistic coefficient, benefit of prey species E to prey species S |
| $\beta_{S,E}$ | 1 | 0 | Competition coefficient, effect of prey species E on prey species S |
| $\mu_E$ | 0.5 | 0.5 | Maximum intrinsic growth rate of prey species E |
| $\mu_S$ | 0.5 | 0.5 | Maximum intrinsic growth rate of prey species S |
| $\gamma_{E,G}$ | 20 | 20 | Burst size of generalist phage on prey species E |
| $\gamma_{E,P}$ | 20 | 20 | Burst size of specialist phage on prey species E |
| $\gamma_{S,P}$ | 20 | 20 | Burst size of specialist phage on prey species S |
| $\varsigma_{E,G}$ | 0.001 | 0.001 | Attachment rate of generalist phage on prey species E |
| $\varsigma_{E,P}$ | 0.001 | 0.001 | Attachment rate of specialist phage on prey species E |
| $\varsigma_{S,P}$ | 0.001 | 0.001 | Attachment rate of specialist phage on prey species E |
| $\delta_E$ | 0.03 | 0.03 | Intrinsic death rate of prey species E |
| $\delta_S$ | 0.03 | 0.03 | Intrinsic death rate of prey species S |
| $\delta_G$ | 0.03 | 0.03 | Intrinsic death rate of generalist phage |
| $\delta_P$ | 0.03 | 0.03 | Intrinsic death rate of specialist phage |
| $\kappa_E$ | 0 | 1 | Half-saturation constant of species E |
| $\kappa_S$ | 0 | 1 | Half-saturation constant of species S |
| $R$ | 2 | 1 | System carrying capacity |

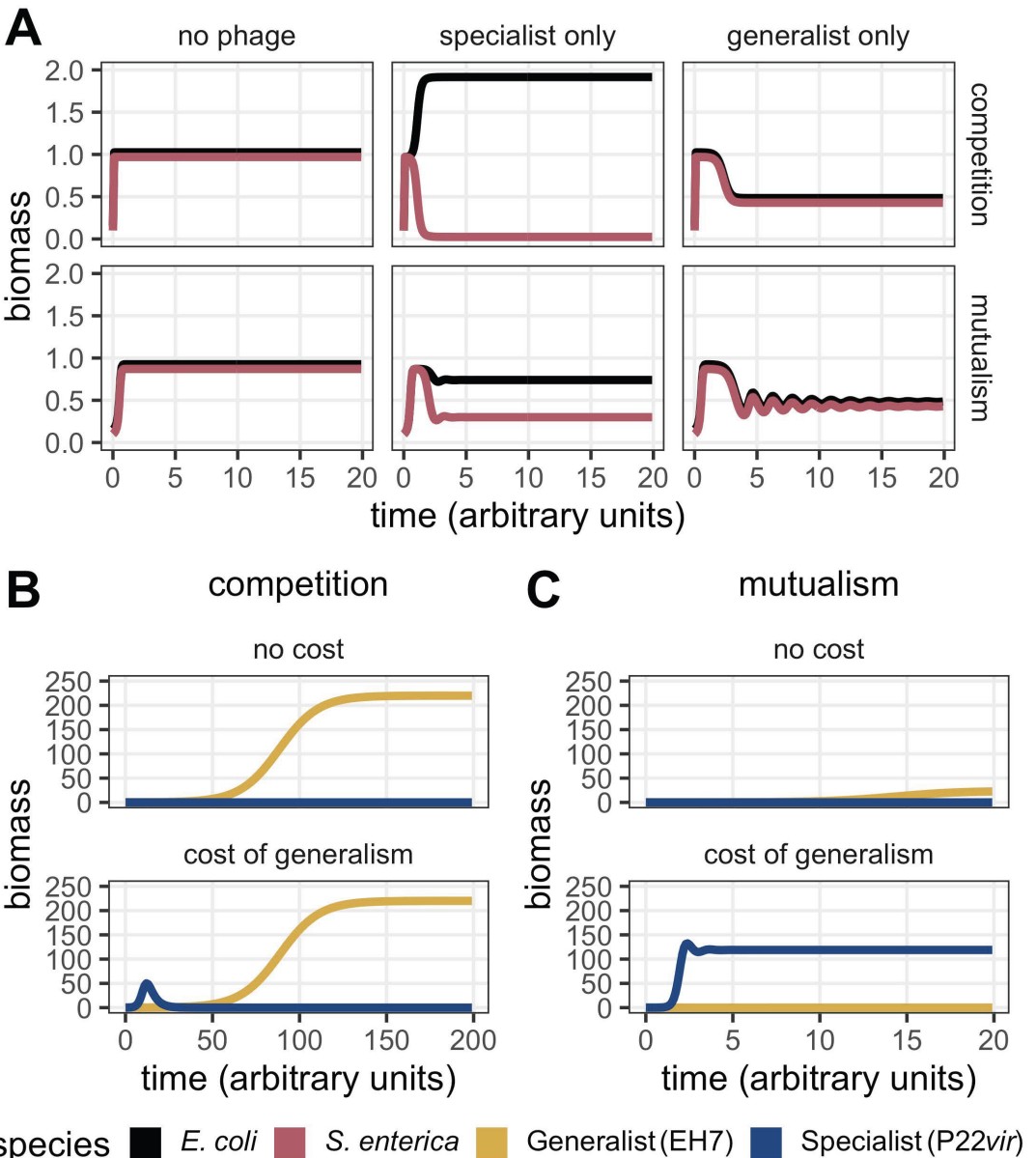

FIG 2 Numerically simulated bacterial dynamics demonstrate that competing prey provide a different selective environment for phage than mutualistic prey. (A) In the absence of phage, both prey species reach an equilibrium ratio of 50:50. In the presence of only a specialist phage, the prey attacked by the specialist (pink line, *Salmonella enterica*) decreases. The prey that is not attacked (black line, *E. coli*) reaches a higher equilibrium frequency if the prey are competing, and a lower equilibrium frequency if the prey are mutualists. When only the generalist predator is present, regardless of prey interactions, prey remain at a 50:50 ratio throughout the simulated growth period. Note that in panels where abundances are identical, biomass has been deliberately offset for effective visualization. (B) When prey compete and both phage types are present and parametrically identical (no cost), the generalist (yellow line, EH7) dominates over the specialist (blue line, P22*vir*). When prey compete and both phage types are present but the specialist's burst size increased to five times that of the generalist (cost of generalism), the generalist (yellow line, EH7) still dominates over the specialist (blue line, P22*vir*). (C) When prey are mutualistic and both phage types are present and parametrically identical (no cost), the generalist (yellow line, EH7) dominates over the specialist (blue line, P22*vir*). When prey are mutualistic and both phage types are present but the specialist's burst size increased to five times that of the generalist (cost of generalism), the specialist (blue line, P22*vir*) dominates over the generalist (yellow line, EH7).

To further understand the behavior of the model when phage phenotypes competed against one another, we sought to simplify the system, applying $R^*$ theory to (equations 2 and 3). $R^*$ theory (33) suggests that, in many cases, when two species compete for the same resource, the species that requires fewer resources for zero net population

growth will be able to competitively exclude the other species. This is formalized by computing the resource needed for zero net population growth, $R^*$, for each species independently and comparing those values. To apply this to our system, we treated the shared prey, *Salmonella enterica*, as a resource, and identified domains in which the amount of *S. enterica* required by the specialist to hold its net population growth at zero, $S_P^*$, would be expected to be less than the amount of *S. enterica* required for zero net growth of the generalist population, $S_G^*$ (33). To do so, we assumed that the generalist phage had the same burst size and attachment rate on both prey types and that the intrinsic rate for mortality due to dilution or death was not species-specific, such that $\gamma_G$ described the burst size of the generalist on both prey, $\varsigma_G$ described the attachment rate of the generalist on both prey, $\gamma_P$ described the burst size of the specialist on *S. enterica*, $\varsigma_P$ described the attachment rate of the specialist on *S. enterica*, and $\delta$ described the intrinsic mortality rate of all four species. $S_G^*$ and $S_P^*$ were obtained by setting the left-hand side of (equations 3 and 4) equal to 0 and solving for $S$.

In accordance with $R^*$ theory, requiring that

$$S_P^* < S_G^* \tag{5}$$

leads to

$$\frac{\gamma_G * \varsigma_G}{\gamma_P * \varsigma_P} < 1 - \frac{\gamma_G * \varsigma_G * E}{\delta} \tag{6}$$

This analysis therefore suggested that the specialist phage could dominate (i.e., have the lower $S^*$) when the alternative prey source *E. coli* was sufficiently rare (*E* sufficiently small), or when the generalist suffered a fitness trade-off for expanded specificity in the form of reduced burst size and/or attachment rate ($\gamma_G * \varsigma_G$ sufficiently small). More generally, a cost of generalism is required for the specialist to dominate, as (equation 5) can only be fulfilled if $\gamma_G * \varsigma_G < \gamma_P * \varsigma_P$. While interactions between prey alone can alter the magnitude of selection, a cost of generalism is required to alter the direction of selection between interaction types. This inequality further supports the relevance of the "jack of all trades, master of none" hypothesis (40–46).

To verify the intuition of our $S^*$ inequality, we used our default parameters (Table 1) to consider various domains in which the specialist phage predator was favored. To do so, we systematically changed the burst size of the specialist phage on the shared prey *S. enterica* ($\gamma_{S,P}$) and considered the relative abundance of each phage type at equilibrium (Fig. 3; for changing attachment rate $\varsigma_{S,P}$ instead of burst size $\gamma_{S,P}$, see Fig. S1). We found that even as the cost of phage generalism increased, the generalist phage was always favored when prey competed, although there were no stable fixed points using our default parameters (Fig. 3A, left panel; File S1). Conversely, the specialist was favored on mutualistic prey (Fig. 2C and 3A, right panel) given a minimum cost of generalism. If $\varsigma_P * \gamma_P > 2.83 \varsigma_G * \gamma_G$, i.e., the product of the specialist's burst size and attachment rate was 2.83 times as large as that of the generalist, then the only stable fixed point was that of the two bacterial species coexisting with the specialist. As such, given a threshold cost of generalism, only the specialist could coexist with mutualistic bacterial species, regardless of initial conditions.

To visualize the next prediction of our $S^*$ inequality—that the relative availability of the alternative prey source *E. coli* mattered for the competitive outcome of phage specificity strategies—we loosened the previous assumption that bacterial species should reach a 50:50 ratio without predators. In addition to varying the ratio of prey growth rates (in both competition and mutualism) or the ratio of interaction coefficients ($\alpha_{E,S}$ for mutualism, $\beta_{E,S}$ for competition), we systematically altered the cost of generalism (either through burst size, in Fig. 3, or attachment rate, in Fig. S1). Our analyses

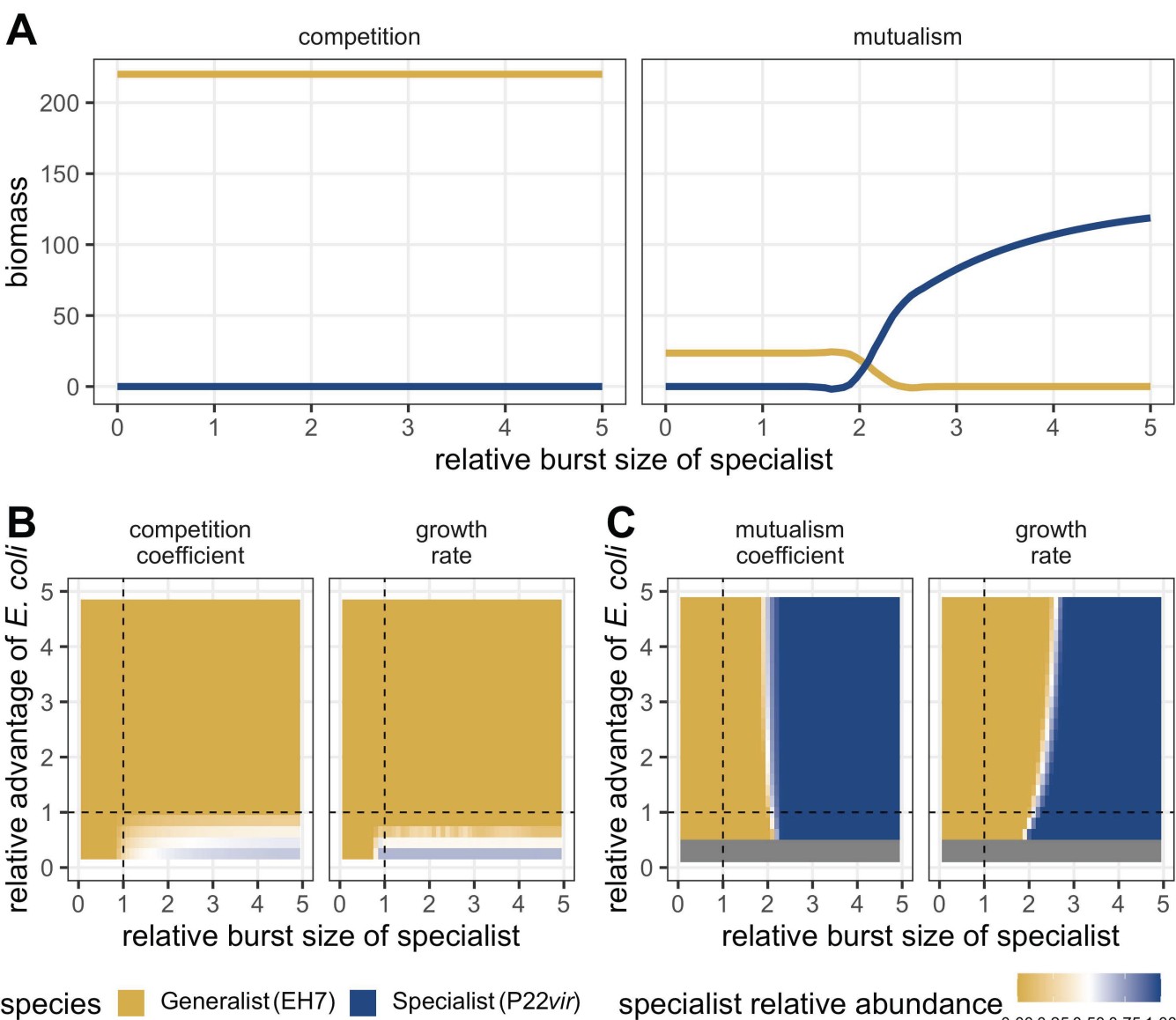

FIG 3 End points of numerically simulated phage dynamics given a variety of parameter trade-offs demonstrate that prey interactions result in different patterns of predator abundance. (A) The final density of each phage type as a function of bacterial interactions and increasing cost of generalism modeled as increasing specialist burst size. When prey are mutualistic, a relative burst size above 2.1 favors specialist phage (blue line, P22*vir*) over generalists (yellow line, EH7). When prey compete, there is no relative burst size that favors the specialist phage; this is true even as the specialist's burst size increases well beyond the values displayed here. For these analyses, the generalist's burst size is set to 20, with the specialist's burst size increased relative to that value. All other default parameter values can be found in Table 1. (B) The relative abundance of the specialist phage on competing prey as a function of increasing cost of generalism and relative growth advantage of the alternative prey *E. coli*. Whether prey growth advantage is modeled through growth rate ($\mu$) or competitive coefficients ($\beta$), the generalist is favored (yellow, EH7) except in a small subset of cases where the alternative prey is competitively excluded. For these analyses, the generalist's burst size is set to 20, with the specialist's burst size increased relative to that value. For the competition coefficient panel, the competition coefficient of *S. enterica* is 1, and the competition coefficient of *E. coli* is set relative to that value. For the growth rate panel, the growth rate of *S. enterica* is 0.5, and the growth rate of *E. coli* is set relative to that value. All other default parameter values can be found in Table 1. (C) The relative abundance of the specialist phage on mutualistic prey as a function of increasing cost of generalism and relative growth advantage of the alternative prey *E. coli*. Whether prey growth advantage is modeled through growth rate ($\mu$) or mutualistic benefit ($\alpha$), a cost of generalism exists above which specialism is favored (blue, P22*vir*). Note that there are benefit and growth rate values for *E. coli* below which the mutualistic system cannot be supported, indicated by the gray bar. For these analyses, the generalist's burst size is set to 20, with the specialist's burst size increased relative to that value. For the mutualism coefficient panel, the mutualism coefficient of *S. enterica* is 1, and the competition coefficient of *E. coli* is set relative to that value. For the growth rate panel, the growth rate of *S. enterica* is 0.5, and the growth rate of *E. coli* is set relative to that value. All other default parameter values can be found in Table 1.

demonstrated that when prey were competing, relative growth rates and competition coefficients mediated the abundance of the specialist such that it had a fitness equivalent to or greater than that of the generalist only in those cases where the alternative prey *E. coli* was competitively excluded by the shared prey *S. enterica* (Fig. 3B). In contrast, when prey were mutualistic, neither relative growth rates nor relative mutualistic benefit coefficients altered phage relative abundance greatly. Instead, ecological dominance of the specialist depended mainly on the cost of generalism, such that the specialist tended to proliferate given a sufficient cost regardless of biased bacterial abundances driven by unequal mutualistic benefit (Fig. 3C).

Finally, to ensure that our tested parameters captured the fundamental behavior of the model, we performed two sensitivity analyses—a Morris screening and a Sobol variance analysis—to determine which parameters had the largest impact on the final biomass of each phage type. In the case of both obligate mutualism and competition, Morris screening methods suggested that the death/dilution rate, burst size and attachment rates of both phage, and the interaction parameters for the microbial species were of greatest impact (Table S1). The variance-based Sobol method reinforced the importance of dilution rate (Table S2). These results were consistent both with the parameters identified by fixed point analysis, our $S^*$ inequality, and the basic construction of the model, which requires that both phage types have reproductive parameters sufficient to offset the chemostat-induced mortality rate.

## Phage relative abundance *in vitro* aligns with modeling results in co-culture

Using our wet-lab experimental cross-feeding system, we tested the mathematical prediction that generalist predators would be favored on competing prey and specialist predators would be favored on mutualistic prey. We first verified that over 48 hours, our specialist phage (P22*vir*) could only replicate on *S. enterica* and our generalist phage (EH7) could replicate on both prey species (Fig. 4A). Each phage alone also grew well on competitive and mutualistic bacterial co-cultures. The final density of EH7 appeared similar when replicating on both interaction types (Fig. 4A, $P = 0.999$; Table S3). This was true for P22*vir* as well, such that type of interaction did not result in significantly different final titers (Fig. 4A, $P = 0.909$; Table S3). However, in those experiments, EH7 reached a higher titer than P22*vir* on both competitive (Fig. 4A, $P < 0.0001$; Table S3) and mutualistic co-cultures (Fig. 4A, $P < 0.0001$; Table S3). Consistent with our model, P22*vir* increased *E. coli* frequency relative to the no-phage control in competitive co-culture (Fig. 4C, $P < 0.0001$; Table S3). Applying the specialist phage suppressed, but did not eliminate, both bacterial species in mutualism (Fig. 4D). In contrast, the generalist phage more effectively suppressed *E. coli* in competitive co-culture, resulting in a higher relative density of *S. enterica* compared to the no-phage control (Fig. 4C, $P = 0.0035$; Table S3). Unlike P22*vir*, the application of EH7 to mutualistic co-culture did not suppress co-culture growth for the duration of the experiment (Fig. 4D).

We also evaluated population dynamics when the two phages competed against one another (Fig. 4B through D). When both phage were present in competitive co-culture, the generalist reached a higher final density than the specialist (Fig. 4B, $P < 0.0001$; Table S3), as predicted by our modeling results. Bacterial dynamics were also consistent with our model, with *E. coli* dominating through competitive release (Fig. 4C, $P < 0.0001$; Table S3). When both phage were present in mutualistic co-culture, the specialist reached a higher final titer (Fig. 4B, $P < 0.0001$; Table S3) and co-culture densities were suppressed for the duration of the experiment (Fig. 4D). Curiously, however, the presence of the specialist phage reduced EH7 below the limit of detection when prey were mutualistic (Fig. 4B), a result that we interrogated further.

## *In vitro,* cost of generalism manifests as increased rate of degradation and reduced infectivity of starved cells

To understand our inability to detect the generalist phage in mutualistic co-culture at the end of our experimental window, we investigated the ability of each phage to reproduce

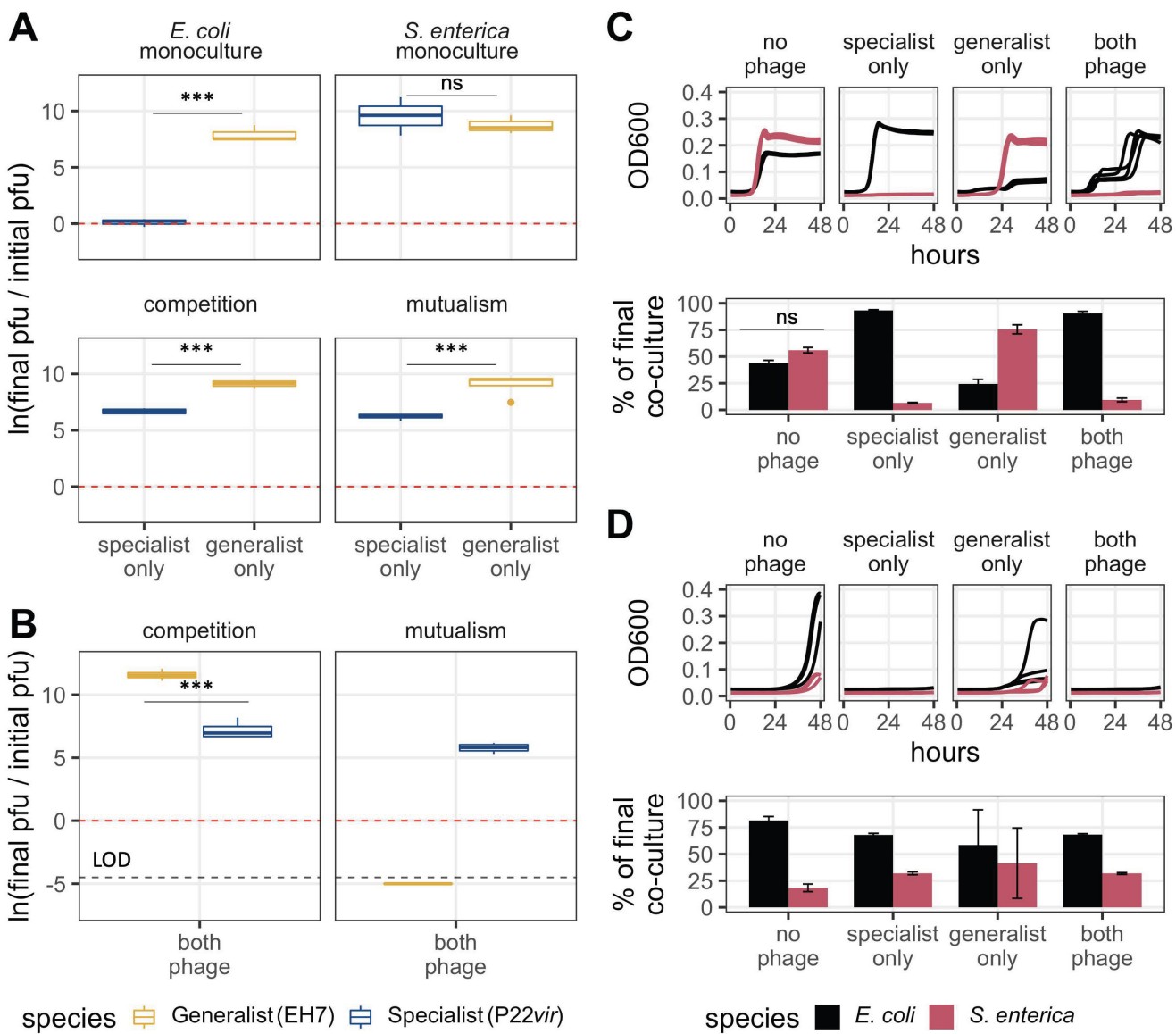

**FIG 4** Phage and bacterial dynamics *in vitro* align with modeling expectations. (A) Change in individual phage titer on bacterial monocultures and co-cultures over a 48-hour time period. EH7 grows on both bacterial strains, while P22*vir* cannot replicate on *E. coli* but does effectively replicate on *S. enterica*. EH7 reaches approximately equivalent densities mixed prey with both interaction types (*P* = 0.999), as does P22*vir* (*P* = 0.909), although its final density is reduced compared to EH7 (competition: *P* < 0.0001, mutualism: *P* < 0.0001). (B) Change in phage titer when phage compete on different bacterial interaction types over a 48-hour time period. When competing against the specialist, EH7 dominates when prey compete (*P* < 0.0001). P22*vir* dominates when prey are mutualistic (*P* < 0.0001), while EH7 disappears below the limit of detection (LOD). For panels A and B: the dotted red line indicates no change in titer from the start of the experiment to the end. Values greater than zero indicate an increase in titer, while values below zero indicate a decrease in titer. Statistical significance was determined using a two-way analysis of variance (ANOVA) with Tukey's honestly significant difference (HSD) multiple comparison test (Table S3). The black dotted line indicates the limit of detection. (C) Inferred species-specific ODs over time across treatment conditions (top) and final fraction of bacterial co-culture composed of each strain across treatment conditions (bottom) when bacteria compete. The fraction of *E. coli* increases in those cases in which *S. enterica* is predominantly suppressed by phage (specialist only and both phage treatments). *E. coli* frequency is reduced relative to no-phage controls when only EH7 is applied (*P* = 0.0035). (D) Inferred species-specific ODs over time across treatment conditions (top) and final fraction of bacterial population composed of each strain across treatment conditions (bottom) when bacteria are mutualistic. *E. coli* dominates at similar levels in all treatment conditions, even in cases where overall growth is suppressed. For panels C and D: statistical significance for bar graphs was determined using a two-way ANOVA with Tukey's HSD multiple comparison test (Table S3). The 600-nm wavelength optical density (OD$_{600}$) traces represent four biological replicates for conditions with phage and three biological replicates for conditions without phage.

on starved cells by adding phage to monocultures in lactose minimal media as described in Materials and Methods. When placed in wells without bacterial cells or with starved *E. coli*, P22*vir* titer remained unchanged over the 48-hour growth period (Fig. 5A). When placed in wells with starved *S. enterica*, which were also expected to be physiologically inaccessible, P22*vir* titer increased relative to the condition without cells (Fig. 5A, *P* = 0.001; Table S4) or with only *E. coli* (Fig. 5A, *P* = 0.0001; Table S4).

In comparison, the generalist phage EH7 decreased in abundance in all conditions after the 48-hour growth period. There were no detectable infectious phage particles

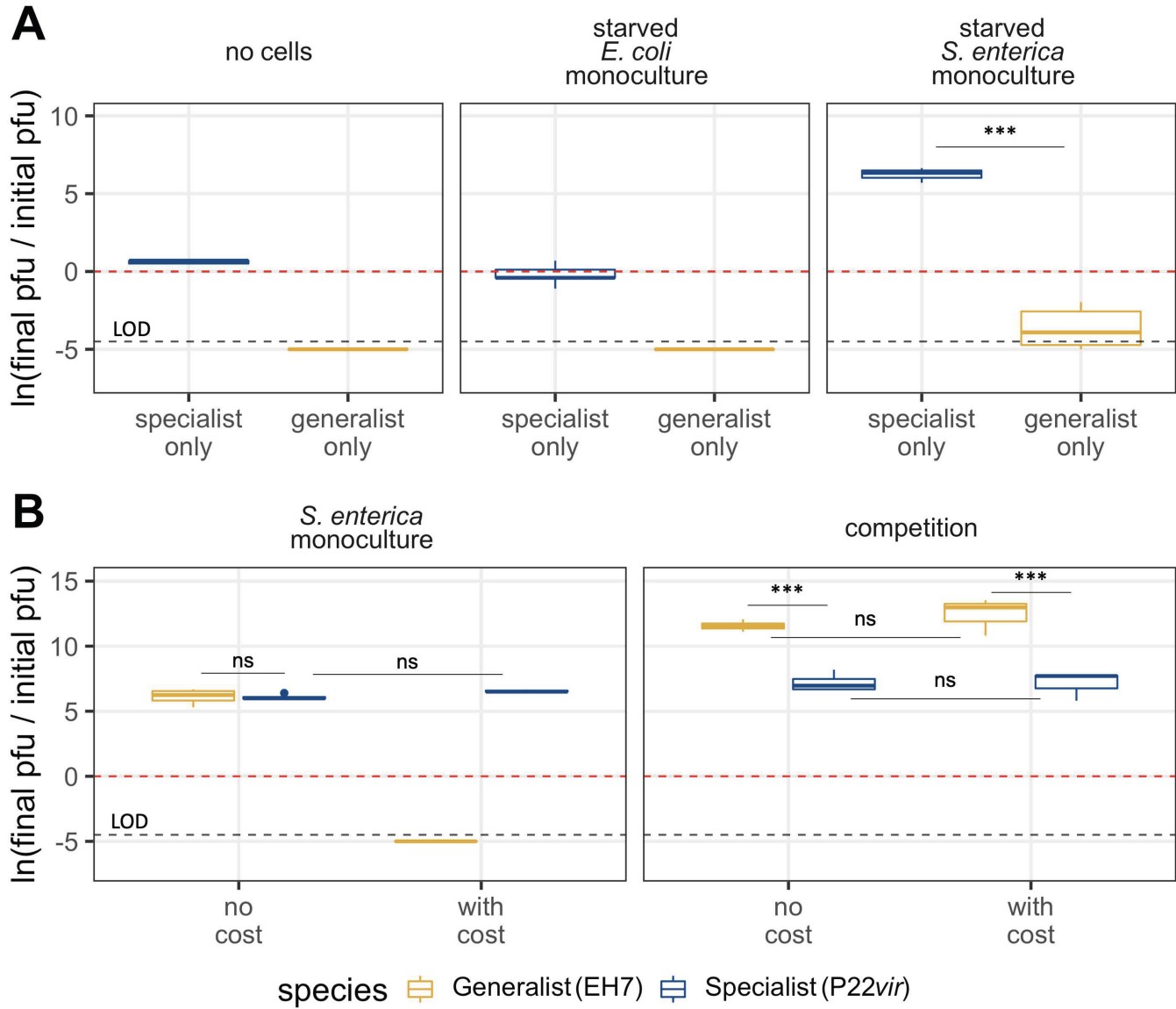

**FIG 5** Imposing a cost of generalism *in vitro* as phage intrinsic mortality reproduces modeling results on *S. enterica* monoculture. (A) Change in phage titer across interaction types and phage treatments when cells are starved or not present in cultures. P22*vir* does not degrade in minimal media over the 48-hour growth period, even in cases when there are no cells present that it can productively infect. Additionally, it can reproduce even on starved *S. enterica*. EH7 degrades below the limit of detection in all conditions but does not disappear completely when *S. enterica* is present. (B) Change in phage titer across interaction types when cells are either added to minimal media at the same time as both phage (no cost) or 24 hours later (with cost). In the no cost condition, EH7 reaches comparable titer to P22*vir* on *S. enterica* monoculture (*P* = 0.999) and wins when prey compete (*P* < 0.0001). In the condition where cells are added after a period of phage incubation, EH7 degrades below the limit of detection on *S. enterica* monoculture but dominates when prey are competing (*P* < 0.0001). For panels A and B: the dotted red line indicates no change in titer from the start of the experiment to the end. Values greater than zero indicate an increase in titer, while values below zero indicate a decrease in titer. Statistical significance was determined using a two-way analysis of variance with Tukey's honestly significant difference (HSD) multiple comparison test (Table S4). The black dotted line indicates the limit of detection (LOD).

in wells without cells or with starved *E. coli*. In wells with only starved *S. enterica*, some infectious phages were still detectable, although phage titer was greatly reduced, suggesting that *S. enterica* may be more physiologically accessible to EH7 than *E. coli* in a starved state (Fig. 5A, *P* = 0.012; Table S4). Taken together, these results indicate that the generalist phage suffers a cost that manifests in two ways: first, a rapid rate of degradation in minimal media, and, second, a reduced capacity to infect and reproduce inside starved cells relative to the specialist phage P22*vir*. We therefore expect that, when competing with P22*vir* on mutualistic co-culture, the starved physiology of the interdependent cells reduces the generalist's infective ability relative to the specialist. This reduction, in combination with the generalist's higher degradation rate, provides an explanation for our inability to detect it at the end of previous experiments.

Interestingly, these results are specific to minimal media, as EH7 does not degrade in LB (Supplemental analysis; Fig. S2; Table S5). We were not able to identify which component of our minimal media was responsible for the degradation of the phage, though it does not appear to be related to the presence of metals or the result of osmolarity (Supplemental analysis; Fig. S2).

## The generalist is favored in competition even when a cost is imposed *in vitro*

The *in vitro* experiments replicated our modeling results when bacterial prey were mutualistic or competing. However, the modeling results assumed a cost of generalism we did not observe in the *in vitro* system when both phage types were competing for actively growing *S. enterica*, with each phage reaching titers that did not differ significantly from one another (Fig. 5B, "no cost," *P* = 0.999; Table S4). A clear cost of generalism may thus only manifest in the mutualistic treatment where the generalist phage EH7 degrades rapidly due to bacterial growth delayed by P22*vir* predation on *S. enterica*. Phage durability is well-established as a critical component of phage fitness and has been shown in some cases to trade off with fecundity (47). This potentially explains our results on the shared prey: while EH7 reproduction on actively growing *S. enterica* matches or exceeds that of P22*vir*, its reproductive capacity may trade off with its degradation rate, given its poorer environmental durability in minimal media.

Therefore, to impose a cost of generalism across all treatments that more closely matched that observed in the mutualistic co-culture, we repeated our phage competition assay experiments by incubating the phage for 24 hours prior to the addition of cells, anticipating that some degradation of the generalist EH7 would occur, while the titer of P22*vir* would remain unchanged. Previous preliminary experiments had indicated that, starting with $10^3$ phage particles, EH7 titer generally drops below the limit of detection after 24 hours (Fig. S2). However, the phage could be recovered following the addition of cells, suggesting that infectious phage particles remained. We expected that the reduced titer of EH7 when cells were applied would mirror the conditions of mutualism when phage competed. Given that the presence of P22*vir* in competitive co-culture increases *E. coli* frequency (Fig. 4C), we hypothesized that a lower titer of EH7 when cells are added may still result in a higher final titer of the generalist phage relative to the specialist, because any remaining EH7 would be able to utilize *E. coli*.

When this cost was imposed and phage were competed on *S. enterica* monoculture, the generalist disappeared below the limit of detection, while P22*vir* final titer was not significantly different than in the condition without cost (Fig. 5B, *P* = 0.982; Table S4). In contrast, we found that when bacteria competed, final titers of both EH7 (Fig. 5B, *P* = 0.723; Table S4) and P22*vir* (Fig. 5B, *P* = 0.999; Table S4) were comparable to the no cost condition, as we had expected. These results are consistent with our modeling results, suggesting that, even with a substantial cost of generalism, a generalist predator should be favored on competing prey.

## In a phenomenological model, prey interactions determine the intrinsic death rate needed to favor specialism

Finally, we amended our model to see if we could replicate the results of our initial *in vitro* experiment, given the known constraints of higher rates of degradation of the generalist phage when cultured in minimal media. To do so, we imposed a cost of generalism not as burst size (or attachment rate) but instead as an increased intrinsic mortality rate for the generalist phage, while keeping the mortality rate the same for the three other species.

We observed that, when fitness cost was modeled as increased mortality, our qualitative results matched those when fitness cost was measured as burst size or attachment rate (compare Fig. 6 with Fig. 3A and Fig. S1A, respectively). The generalist phage could be maintained on competing prey even as its mortality rate increased significantly; competition with the specialist phage decreased the abundance of the generalist, but competitive release of *E. coli* ensured that the generalist had access to sufficient prey for positive population growth across a wide range of parameters (Fig. 6). In contrast, the same intrinsic mortality rate at which the generalist persisted on competing hosts (Fig. 6) resulted in the loss of the generalist when prey were mutualistic due to the specialist phage driving correlated reductions of both *S. enterica* and *E. coli* (Fig. 6). It is worth noting that our model is not currently designed to capture changes in the relative phage attachment rates or burst sizes as a function of physiological state. Therefore, our model assumes that the relative reproductive abilities of

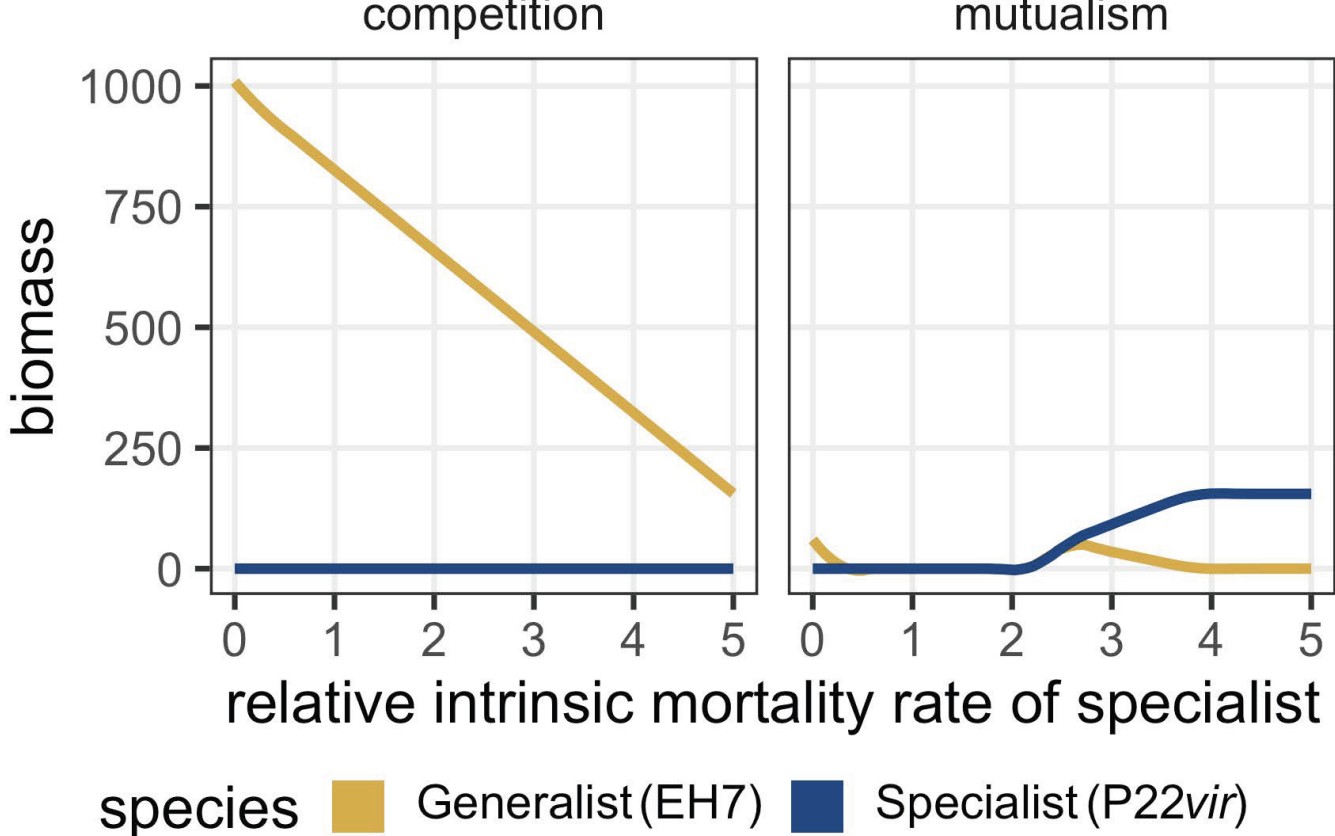

**FIG 6** When a cost of generalism is modeled as intrinsic mortality, qualitative patterns of ecological selection on predator specificity match findings when cost of generalism is modeled as burst size (Fig. 3) or attachment rate (Fig. S1). As the intrinsic mortality rate of the generalist phage increases, it maintains its advantage longer when prey compete and is driven extinct at a minimal cost when prey are mutualistic. These qualitative results align with previous modeling findings when cost of generalism is imposed as burst size or attachment rate. For these analyses, the intrinsic mortality rate of the specialist was set to 0.0067, with the generalist's mortality rate increased relative to that value. All other default parameter values can be found in Table 1.

the two phage are static and that the relationship between phage reproductive rate and intrinsic mortality is driven solely by the relative abundance of prey, assumptions which do not capture the full biological reality of our *in vitro* system. Regardless, in its current form, it reinforces our previous qualitative modeling predictions and our *in vitro* findings that suggest that, when a cost of generalism is present, regardless of its mechanism, competition between prey favors generalist predators across even severe fitness trade-offs, while a minimal fitness cost favors specialist predators when prey are engaged in mutualism.

## DISCUSSION

We aimed to determine whether ecological interactions between bacterial prey species impacted the abundance of phage with different specificities. We developed a simple four-species phenomenological model composed of two interacting bacterial species, a specialist phage, and a generalist phage. Using this chemostatic model, given a cost of generalism, we found that specialist phage were favored when prey are mutualistic, while generalist phage were favored when prey compete. These qualitative results were largely robust to initial conditions, suggesting that they may be both ecologically and evolutionarily informative. We found that our modeling predictions were well-matched by the outcome of batch culture experimental phage competition assays. The alignment between outcomes in our model and our system was observed despite differences in the mechanisms driving the cost of generalism. In our model, a cost of generalism was imposed as either a lower burst size or worse attachment rate and was assumed to be static over time. *In vitro*, the cost of generalism was dynamic over time, driven by differences in phage durability and ability to reproduce on bacteria in variable physiological states. Our results highlight that ecological interactions between prey can alter ecological selection on predator specificity in predictable ways when a cost of generalism exists, regardless of the exact mechanism of cost.

Our modeling results suggest that interactions between bacterial prey impact the prevalence of phage specificity phenotypes when a cost of generalism exists. Experimental evolution has previously shown that the presence of different types of resources can select for generalism (6, 19, 40). Both absolute and relative prey densities are relevant predictors of phage specificity (22, 23, 27, 28). However, while much of the previous work done on diet breadth has assumed a constant relative abundance of available prey, our model upended that assumption by allowing relative prey abundances to vary as a function of prey ecology. Previous theoretical modeling has demonstrated that resource competition between prey species can select for expanded predator diet breadth even when trade-offs for generalism exist, although this result generally required the competitive dominance of the novel prey source (29, 34). Our results align with these findings, underscoring that resource competition should favor a generalist strategy in most cases, even when a severe fitness trade-off is present. Additionally, we expanded previous findings to include mutualistic interactions between prey, showing that a specialist predator strategy dominated assuming even a minimal trade-off for generalism. Our model demonstrates that ecological interactions between prey species favor different predator diet strategies when there is a cost of generalism because switching from competition to mutualism changes relative prey abundances from being anti-correlated to being positively correlated. We anticipate that our modeling result will apply to systems when interactions between prey generate correlations in their abundance and a cost of generalism is present.

The experimental results of our study align with our modeling predictions, although they also highlight two important aspects of our microbial system. First, *in vitro*, we did not observe a reproductive fitness cost of generalism on the shared prey species *S. enterica* under optimal growth conditions, a cost we had anticipated in our model. Instead, we found that the trade-off manifested as an interplay between the generalist's increased degradation in minimal media and its reduced replication rates on starved cells. Our results contribute to the body of work suggesting that pleiotropic costs

are often context dependent (18, 36, 40–46). They also reinforce the observed importance of durability as an important component of phage fitness (47). Additionally, our experiments emphasize that in addition to altering population dynamics, interactions between bacteria can impact phage specificity by altering prey physiology. Bacterial sensitivity to phage can change between physiological states due to differences in growth rate, metabolism, transcription and translational activity, and the availability of various intracellular components (48–50). Phage reproduction on slow-growing or stationary phase cells is often more difficult due to reduced cell size and lower densities of the receptors phage use to adsorb (28, 51). Because microbial physiology is driven by resource availability and interactions between bacterial species (48), our work suggests that ecological selection on phage specificity is impacted by how interactions between bacterial prey shift prey abundances and physiological states over time.

There are limitations to the study we performed that may impact the generality of the results. First of all, our study is limited by its focus on the types of interactions that we chose to examine, namely obligate cross-feeding and resource competition. Other interactions or even other types of mutualistic or competitive interactions—for example, defensive mutualisms or interference competition—could result in different selective patterns on phage diet breadth. Additionally, we do not consider bacterial resistance to phage, which would create subpopulations within interacting species and complicate correlations in bacterial abundance.

We also note that the two phage types tested in these experiments differ in ways unrelated to specificity. P22*vir* attaches to the O-antigen of *S. enterica*'s lipopolysaccharides (LPS), while EH7 uses BtuB, a vitamin B12 uptake receptor. Previous work has demonstrated that BtuB expression is context dependent, while the LPS is constitutively expressed (52–60). Our phages are also different sizes, with EH7 consisting of 110 kb and 154 putative proteins, while P22*vir* is much smaller at 41 kb and 72 proteins (61). We expect that the poor infective capacity of EH7 on starved cells is a result of both receptor type and phage size. When cells are starved, reduced BtuB receptor density may depress EH7 attachment rate, while P22*vir* attachment rate remains unchanged. Likewise, differences in phage genome size and protein content mean that intracellular demands for producing EH7 will be higher than P22*vir*, so its burst size may be reduced when cells are starved. Critically, EH7 is also less durable than P22*vir* in minimal media. However, these differences, while notable, are present whether phage are grown on competitive or mutualistic co-cultures; bacterial interactions determine the consequences of these differences for phage fitness. Fundamentally, the differences between the two phage types ensure that there is a cost for the generalist phage in our *in vitro* system—though the cost may not be directly due to generalism itself—and our work suggests that, if a cost of generalism exists in some form, bacterial interactions will have consequences for the direction of ecological selection on phage host range. Future work should test the competitive ability of phages that use the same or similar receptors and with greater stoichiometric similarity, though we expect that, in cases where a cost of generalism exists, our results will be applicable.

Our results suggest numerous directions for future study. It would be interesting to select EH7 for increased durability in minimal media to examine whether the improvement is sufficient to offset reproductive costs on slow-growing cells, or if a trade-off in fecundity is observed (47). In the context of phage therapy, the performance of EH7 in minimal media emphasizes the necessity of testing how different environments affect phages and whether phage characteristics such as specificity tend to correlate with susceptibility to degradation (62). These data also suggest that the ways bacteria modify their environments through alteration to local pH or metabolite concentrations will have consequences for their viral predators. For example, human gut microbes sometimes compete with their hosts for vitamin B12 (63); the resultant availability of B12 in the human gut may alter the efficacy of BtuB-specific phages in phage therapy applications. Continued characterization of phage-bacteria interactions in the complex communities in which they are found will improve our ability to use phage for engineering and

biomedical purposes. We also expect that increasing the number of bacterial species or incorporating the evolution of resistance will complicate our findings by allowing for the emergence of phage with intermediate specificities. Finally, we note that the spatial structure of interacting bacterial species, as in a biofilm, will alter local prey availability in natural environments such that our results may not be applicable (38).

We took a simple modeling approach, paired with an ecological experiment, to gain insight into the role of prey ecology on the competitive ability of bacteriophage with different specificities. We found that, in both our model and *in vitro* experiments, prey interactions shaped the prevalence of phage specificity phenotypes, though the focal mechanisms differed between our modeling approach and synthetic community. Management and design of microbial communities is contingent upon our ability to predict the evolutionary outcomes and higher-order ecological effects of multitrophic interactions. Understanding the complex biotic factors driving ecological and evolutionary outcomes for bacteriophage is a critical step for harnessing microbes in industrial and biomedical applications. We suggest that microbial interactions should be studied across a diversity of systems to understand the generality of their impact on phage host range.

## MATERIALS AND METHODS

### Model description

We constructed a model of the concentrations of two interacting bacterial species, a generalist phage, and a specialist phage. Bacteria (dimensionless biomass denoted by $E$ or $S$) can either compete for resources or engage in obligate mutualism (adapted from Hoek et al. [64]). Biomass of prey changes through growth (with Lotka-Volterra-like dynamics depending on the interaction with other prey species) and decreases due to predation and death or dilution:

$$\frac{dE}{dt} = \mu_E * E * \frac{\alpha_{E,S} * S}{(\alpha_{E,S} * S) + \kappa_E} * (R - E - (\beta_{E,S} * S)) - (\varsigma_{E,G} * G * E) - (\delta_E * E) \quad (1)$$

$$\frac{dS}{dt} = \mu_S * S * \frac{\alpha_{S,E} * E}{(\alpha_{S,E} * E) + \kappa_S} * (R - S - (\beta_{S,E} * E)) - (\varsigma_{S,P} * P * S) - (\varsigma_{S,G} * G * S) - (\delta_S * S) \quad (2)$$

Biomass of predators increases through predation and decreases through death or dilution:

$$\frac{dG}{dt} = (\gamma_{S,G} * \varsigma_{S,G} * G * S) + (\gamma_{E,G} * \varsigma_{E,G} * G * E) - (\delta_G * G) \quad (3)$$

$$\frac{dP}{dt} = (\gamma_{S,P} * \varsigma_{S,P} * P * S) - (\delta_P * P) \quad (4)$$

Our model was constructed such that prey had an intrinsic maximum growth rate $\mu_i$, carrying capacity $R$, and three parameters determined prey interactions: $\kappa_i$, $\alpha_{j,i}$, and $\beta_{j,i}$. Mutualism is determined by the saturation constant $\kappa_i$ and the mutualism coefficient $\alpha_{j,i}$, which reflects the beneficial effect of prey species $i$ on the per capita growth of prey species $j$. If all $\kappa_i$ and $\alpha_{j,i}$ values are positive, bacterial species grow faster together and cannot grow alone. Competition is driven by the coefficient $\beta_{j,i}$, where $\beta_{j,i}$ determines the competitive effect of prey species $i$ on the per capita growth rate of prey species $j$. Phage reproduction is modeled via adsorption with attachment rate on species $i$ by phage X as $\varsigma_{i,X}$ which directly leads to lysis by phage X on species $i$ with burst size $\gamma_{i,X}$. The default natural death rate $\delta_i$ was initially identical for all four species, as in a chemostat (Table 1), unless noted otherwise.

Using these equations, we investigated the extremes of pure mutualism ($\kappa_i$ and $\alpha_{j,i} > 0$, $\beta_{j,i} = 0$) and pure competition ($\kappa_i = 0$, $\alpha_{j,i} = 1$, $\beta_{j,i} > 0$) (Table 1). Carrying capacity was

increased when prey competed relative to the value used when prey were mutualistic in order to standardize the total final biomass in each condition. This is required by a structural limitation of the Lotka-Volterra style model, where obligate mutualism inflates the carrying capacity over that set by the value of $R$.

## Model analyses

To predict the biomass of both prey and predator over time, we numerically solved our ordinary differential (equations 1–4) (ODEs) in R v.4.2.1 with the DeSolve package v.1.32, using the LSODA solver. To investigate the equilibrium or steady-state dynamics of the system of equations, we integrated (equations 1–4) until species abundances no longer changed between timepoints. These results were verified by fixed point stability analysis in Mathematica 13.2.1. We evaluated equilibrium abundance of the phage predators under three different scenarios: (i) imposing a trade-off for expanded specificity by penalizing the burst size (or attachment rate) of the generalist phage, (ii) altering the intrinsic growth rates or interaction coefficients of the bacterial prey, or (iii) some combination of scenarios i and ii (Table 2). To quantify phage coexistence, we used the equilibrium abundance of both phage. Relative abundance was calculated as the equilibrium density of the specialist divided by the sum of the equilibrium density of the

**TABLE 2** Parameter trade-offs tested in our phenomenological model and their biological significance[a]

| Trade-off | Parameter combinations | Significance |
|---|---|---|
| None | $\gamma_{S,G} = \gamma_{S,P} = 20$ and $\varsigma_{S,G} = \varsigma_{S,P} = 0.001$ | Generalist and specialist phage are parametrically identical |
| Cost of generalism (burst size) | $\gamma_{S,G} = \gamma_{E,G} = 20$ and $\gamma_{S,G} < \gamma_{S,P}$ | Generalist and specialist phage differ in their abilities to kill prey due to differences in burst size |
| Cost of generalism (attachment rate) | $\varsigma_{S,G} = \varsigma_{E,G} = 0.001$ and $\varsigma_{S,G} < \varsigma_{S,P}$ | Generalist and specialist phage differ in their abilities to kill prey due to differences in attachment rate |
| Interaction outcome (growth rate) | $\mu_S = 0.5$ and $\mu_E \neq \mu_S$ | Prey species coexistence in the absence of phage is biased or impossible due to differences in growth rate |
| Interaction outcome (interaction coefficient, competition) | $\beta_{S,E} = 1$ and $\beta_{E,S} \neq \beta_{S,E}$ | Prey species coexistence when competing in the absence of phage is biased or impossible due to differences in interaction coefficients |
| Interaction outcome (interaction coefficient, mutualism) | $\alpha_{S,E} = 1$ and $\alpha_{E,S} \neq \alpha_{S,E}$ | Prey species coexistence when mutualistic in the absence of phage is biased or impossible due to differences in interaction coefficients |
| Cost of generalism and interaction outcome (growth rate) | ($\gamma_{S,G} = 20$ and $\gamma_{S,G} < \gamma_{S,P}$ and $\mu_S = 0.5$ and $\mu_E \neq \mu_S$) or ($\varsigma_{S,G} = 0.001$ and $\varsigma_{S,G} < \varsigma_{S,P}$ and $\mu_S = 0.5$ and $\mu_E \neq \mu_S$) | Generalist and specialist phage differ in their ability to kill prey and prey species coexistence in the absence of phage is biased or impossible due to differences in growth rate |
| Cost of generalism and interaction outcome (interaction coefficient, competition) | ($\gamma_{S,G} = 20$ and $\gamma_{S,G} < \gamma_{S,P}$ and $\beta_{S,E} = 1$ and $\beta_{E,S} \neq \beta_{S,E}$) or ($\varsigma_{S,G} = 0.001$ and $\varsigma_{S,G} < \varsigma_{S,P}$ and $\beta_{S,E} = 1$ and $\beta_{E,S} \neq \beta_{S,E}$) | Generalist and specialist phage differ in their ability to kill prey and prey species coexistence when competing in the absence of phage is biased or impossible due to differences in interaction coefficients |
| Cost of generalism and interaction outcome (interaction coefficient, mutualism) | ($\gamma_{S,G} = 20$ and $\gamma_{S,G} < \gamma_{S,P}$ and $\alpha_{S,E} = 1$ and $\alpha_{E,S} \neq \alpha_{S,E}$) or ($\varsigma_{S,G} = 0.001$ and $\varsigma_{S,G} < \varsigma_{S,P}$ and $\alpha_{S,E} = 1$ and $\alpha_{E,S} \neq \alpha_{S,E}$) | Generalist and specialist phage differ in their ability to kill prey and prey species coexistence when mutualistic in the absence of phage is biased or impossible due to differences in interaction coefficients |

[a]Parameters not listed here were set to default values in Table 1.

specialist plus the equilibrium density of the generalist. Values greater than 0.5 indicated that the specialist was more abundant. Initial densities were the same across numerical simulations; all four species were always initialized at a density of 0.1. To confirm the significance of the parameters tested, we conducted two types of sensitivity analyses on our ODE system: the Morris screening method and the variance-based Sobol test (65–67). Morris screening and Sobol sensitivity analyses were performed in R with the ODESensitivity package v.1.1.2 using the same parameter distribution ranges for each test type (Table S6).

Finally, following the *in vitro* finding of high rates of degradation of the generalist phage, we amended our model to impose a cost of generalism by increasing the death rate of the generalist relative to the three other species.

## Bacterial co-culture system and phage strains

The bacterial strains have been previously described (68). Strains are listed in Table S7. Our *Salmonella enterica* serovar Typhimurium LT2 strain secretes methionine due to mutations in *metA* and *metJ* (69). The *Escherichia coli* is a methionine auxotroph due to a deletion of *metB* (68). To track bacterial abundances and relative ratios during growth, *E. coli* was tagged with a cyan fluorescent protein (CFP) and *S. enterica* was tagged with a yellow fluorescent protein (YFP) (70).

The specialist phage used was P22*vir*. It is an obligately lytic version of the lysogenic *S. enterica*-specific phage P22, created through several point mutations in its prophage repressor gene (Table S8). P22*vir* was provided by I. J. Molineux. A generalist phage strain, EH7, was isolated and provided by E. Hansen and S. Bowden. EH7 is an obligately lytic T5-like siphovirus that uses BtuB, a differentially expressed outer membrane protein for vitamin B12 uptake, as a receptor. It is similar to T5-like coliphages described in Kim and Ryu (16) and Switt et al. (61).

Two additional bacterial strains were used for plaque assays (Table S7). They were chosen so that, in mixed cultures of phage, phage types could be quantified independently of each other. The *E. coli* K-12 BW25113 Δ*trxA* from the Keio collection was used to quantify EH7 densities as no plaques of P22*vir* form on that host (71). An *S. enterica* serovar Typhimurium NCTC 74 strain with *btuB* knocked out through a transposon insertion (EZ-Tn5 <Kan-2>, Lucigen) was used to quantify P22*vir* as no plaques of EH7 form on that host. The Δ*btuB S. enterica* strain was provided by S. Bowden.

## Media

Minimal hypho liquid media for experiments was prepared as previously described, with each component sterilized prior to mixing (72) (Table S9). In addition to the appropriate carbon source, solutions containing sulfur, nitrogen, phosphorus, and metals were supplemented into each media type (Table S9). Routine culturing of all bacterial strains was carried out on Miller lysogeny broth (LB) unless otherwise indicated. Working stocks of both phage types were grown on log-phase *S. enterica* LT2 cultures in LB and stored at 4°C. Stock titer was determined by plaque overlay assay on the appropriate strains.

## Phage competition assays

Phage competition assays were performed in 96-well flat bottom plates on a Tecan Infinite Pro200 plate reader for 48 hours at 37°C with shaking at 432 rotations per minute. Experiment duration was chosen to allow batch culture experiments to reach a final state (stationary phase, phage densities unchanging), thus allowing us to compare to our chemostatic model. Overnight stationary phase cultures in LB started from single colonies were washed three times in saline, adjusted to a density of $10^7$ cells per milliliter, and used to inoculate 200 µL of appropriate medium with $2.0 \times 10^5$ total cells per well (i.e., $2.0 \times 10^5$ total *S. enterica* cells in monoculture, $1.0 \times 10^5$ total *S. enterica* cells, and $1.0 \times 10^5$ total *E. coli* cells in co-cultures). Phage stocks were diluted in saline to $10^5$ plaque-forming units (PFUs) per milliliter and inoculated into the appropriate wells to a

multiplicity of infection (MOI) between 0.005 and 0.01 for a given phage, depending on the fraction of infectable cells for each phage type. Phage strains were added either in isolation ($10^3$ total phage particles of either P22*vir* or EH7) or in a one-to-one ratio ($10^3$ total phage particles of P22*vir* and $10^3$ total phage particles of EH7 for a final density of $2 \times 10^3$ total phage particles). Phage densities were confirmed by plaque overlay assay at the start of the experiment on the appropriate strains.

To quantify bacterial abundances throughout the duration of the experiment, we recorded 600-nm wavelength optical density ($OD_{600}$), *E. coli*-specific CFP (Ex: 430 nm; Em: 480 nm), and *S. enterica*-specific YFP (Ex: 500 nm; Em: 530 nm) fluorescence every 20 minutes. We converted fluorescent protein signals to species-specific OD equivalents as previously described (73). To determine phage density at the end of the 48-hour growth period, we plated for PFUs for each replicate from half of the total 200 µL volume using plaque overlay assays on LB plates with 0.7% LB top agar. All replicates were quantified using plaque assays on both Δ*trxA E. coli* and Δ*btuB S. enterica*. Δ*trxA E. coli* and Δ*btuB S. enterica* were prepared for use in plaque assays through overnight culture growth in LB, prior to being diluted 1:10 (Δ*btuB S. enterica*) or 1:5 (Δ*trxA E. coli*) and allowed to grow for 30 minutes. Plaque assays were otherwise performed as previously described using 2 µL of phage spot dilutions from $10^0$ to $10^{-7}$ with three technical replicates per dilution per sample (51, 74). The lower limit of detection was 500 PFU/mL. Change in phage titer was represented as the natural log of the final phage density divided by the starting phage density [ln (final PFU/mL / initial PFU/mL)]. All plates were incubated overnight at 37°C.

A single initial experiment was completed to confirm the reproductive ability of each phage on *S. enterica* monoculture or *E. coli* monoculture. Full factorial experiments testing all three phage conditions (P22*vir*, EH7, or EH7 + P22*vir*) on either *S. enterica* monoculture, mutualistic co-culture, or competitive co-culture were then completed with four biological replicates per condition, plus three biological replicates for no-phage controls per condition. Three independent experiments were set up and completed during different weeks to protect against batch effects and confirm the repeatability of the results. One representative experimental run was chosen for display in this paper.

To impose a cost of generalism in our system, we repeated the phage competition assays, incubating the phage in minimal media at 37°C with shaking for 24 hours prior to the addition of cells in either *S. enterica* monoculture or competitive co-culture. Mutualistic co-culture was not tested. Once cells were added, cultures were grown for an additional 24 hours. Phage densities were quantified at the beginning and end of the 48-hour experiment. The experiment was completed once following preliminary trials to confirm that EH7 did not degrade below the limit of recovery after 24 hours.

## Phage degradation assays

We examined the impact of cell starvation on the formation of new EH7 particles using a full factorial design of both phage types and *E. coli* or *S. enterica* monoculture in lactose hypho minimal media. Neither bacterial strain could grow, as each was starved of essential nutrients. Bacteria were inoculated in lactose hypho monoculture at a density of $10^5$ cells per 200 µL of medium in a 96-well plate and treated with either EH7 or P22*vir* at a total density of $10^3$ PFU (MOI = 0.01) or incubated in a no-phage control. Additionally, we tested each phage in isolation in lactose minimal media without cells to determine phage decay rates. Each condition consisted of three technical replicates. Both experiments were completed once at 37°C with shaking at 432 rotations per minute. Phage density in each well was determined by plaque overlay assay at the beginning and end of the 48-hour experiment. Change in phage titer was again represented as the natural log of the final phage density divided by the starting phage density [ln (final PFU/mL/initial PFU/mL)].

## Phage sequencing and genomic analysis

Phage samples were sequenced following dsDNA extraction. To isolate DNA, 450 µL of each phage stock was combined in a microcentrifuge tube with 50 µL DNase I 10× buffer

(Invitrogen), 5 µL DNase I (Invitrogen), and 1 µL RNase A (Qiagen). The solution was incubated at 37°C without shaking for 1.5 hours, followed by inactivation of DNase I and RNase A through the application of 20 µL of 0.5M EDTA and incubation at 75°C for 10 minutes. Proteinase K (1.25 µL; Invitrogen) was then added to the tube and the solution was incubated for an additional 1.5 hours at 56°C without shaking. DNA was purified using the Qiagen DNeasy Blood and Tissue kit and quantified on a Nanodrop. Samples were sent to Seq Center, LLC (https://www.seqcenter.com/), for sequencing.

Once returned, reads were assembled and evaluated. Point mutations knocking out lysogeny in our lab strain of P22*vir* (reads available at Sequence Read Archive [SRA] accession SRX22993822) were identified using breseq v.0.28 (75) to assemble and analyze reads relative to the ancestral, lysogenic version of P22 (GenBank accession NC_002371.2). A complete EH7 annotation was created by E. Hansen and S. Bowden using Unicycler v.0.5.0 (76) (GenBank accession OR413347.1).

## ACKNOWLEDGMENTS

The authors thank X. Xiong, J. N. V. Martinson, J. Chacón, A. K. Shaw, M. Torstenson, C. Wojan, N. Narayanan, and D. Kim for helpful comments on the manuscript. The authors would also like to thank members of the Möbius and Nadell labs for input throughout the conceptualization and design of this project, as well as I. J. Molineux for providing our lytic P22*vir* strain.

This work was supported by BBSRC via BBSRC-NSF/BIO grant BB/V011464/1 to W.M. and the National Science Foundation IOS-2017879 to C.D.N. and IOS-2019304 to W.R.H.

## AUTHOR AFFILIATIONS

[1]Department of Ecology, Evolution and Behavior, University of Minnesota, St. Paul, Minnesota, USA
[2]Living Systems Institute, University of Exeter, Exeter, United Kingdom
[3]Department of Physics and Astronomy, University of Exeter, Exeter, United Kingdom
[4]Department of Biological Sciences, Dartmouth College, Hanover, New Hampshire, USA
[5]Department of Food Science and Nutrition, University of Minnesota, St. Paul, Minnesota, USA
[6]BioTechnology Institute, University of Minnesota, St. Paul, Minnesota, USA

## AUTHOR ORCIDs

Ave T. Bisesi http://orcid.org/0000-0001-9076-7384
Wolfram Möbius http://orcid.org/0000-0003-0926-4590
Carey D. Nadell http://orcid.org/0000-0003-1751-4895
Steven D. Bowden http://orcid.org/0000-0001-5438-2532
William R. Harcombe http://orcid.org/0000-0001-8445-2052

## FUNDING

| Funder | Grant(s) | Author(s) |
| --- | --- | --- |
| National Science Foundation (NSF) | IOS-2019304 | William R. Harcombe |
| National Science Foundation (NSF) | IOS-2017879 | Carey D. Nadell |
| UKRI \| Biotechnology and Biological Sciences Research Council (BBSRC) | BB/V011464/1 | Wolfram Möbius |

## AUTHOR CONTRIBUTIONS

Ave T. Bisesi, Conceptualization, Data curation, Formal analysis, Investigation, Methodology, Project administration, Visualization, Writing – original draft, Writing – review and editing | Wolfram Möbius, Conceptualization, Funding acquisition, Writing – review

and editing | Carey D. Nadell, Conceptualization, Funding acquisition, Writing – review and editing | Eleanore G. Hansen, Resources, Writing – review and editing | Steven D. Bowden, Resources, Writing – review and editing | William R. Harcombe, Conceptualization, Funding acquisition, Supervision, Writing – original draft, Writing – review and editing

## DATA AVAILABILITY

Numerical simulations, sensitivity analyses, data analysis, statistics, and figure generation were performed using R v.4.2.1 using custom scripts available at https://github.com/bisesi/Host-Ecology-and-Host-Range. Raw experimental data and Mathematica notebooks for fixed point analysis are available at the same link.

## ADDITIONAL FILES

The following material is available online.

### Supplemental Material

**File S1 (mSystems01177-23-s0001.pdf).** Supplemental fixed point analysis.
**Supplemental material (mSystems01177-23-s0002.docx).** Supplemental analysis, figures, and tables.
**Table S1 (mSystems01177-23-s0003.xlsx).** Morris screening global indices of each ODE parameter on generalist or specialist predator density.
**Table S2 (mSystems01177-23-s0004.xlsx).** First order (S) and total effect (T) Sobol sensitivity indices for each ODE parameter on generalist or specialist predator density.
**Table S3 (mSystems01177-23-s0005.xlsx).** Statistical analysis associated with Figure 4.
**Table S4 (mSystems01177-23-s0006.xlsx).** Statistical analysis associated with Figure 5.
**Table S5 (mSystems01177-23-s0007.xlsx).** Statistical analysis associated with Figure S2.

### Open Peer Review

**PEER REVIEW HISTORY (review-history.pdf).** An accounting of the reviewer comments and feedback.

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
