## [Reviewer comments · mSystems]

Bacteriophage specificity is impacted by interactions between bacteria

Ave Bisesi, Wolfram Möbius, Carey Nadell, Eleanore Hansen, Steven Bowden, and William Harcombe

Corresponding Author(s): William Harcombe, University of Minnesota Twin Cities

Review Timeline:

Submission Date:	November 7, 2023
Editorial Decision:	December 14, 2023
Revision Received:	January 8, 2024
Editorial Decision:	January 15, 2024
Revision Received:	January 17, 2024
Accepted:	January 20, 2024

Editor: Zoe Dyson

Reviewer(s): Disclosure of reviewer identity is with reference to reviewer comments included in decision letter(s). The following individuals involved in review of your submission have agreed to reveal their identity: Małgorzata Barbara Łobocka (Reviewer #2)

Transaction Report:

DOI: <https://doi.org/10.1128/msystems.01177-23>

Re: mSystems01177-23 (Bacteriophage specificity is impacted by interactions between bacteria)

Dear Dr. William R Harcombe:

- Please include GenBank/ENA accession numbers for the genome sequences of phages EH7 and P22vir detailed in supplementary tables 6-7.
- Please also include methods details for data presented in supplementary tables 6-7.
- Please also ensure all references listed are included in the manuscript text. For example, I was unable to find call outs to references 49 & 50.
- Please consider the suggestions provided by reviewer #1 (detailed below).

Revision Guidelines

Sincerely,
Zoe Dyson
Editor
mSystems

Reviewer #1 (Comments for the Author):

I would like to thank the authors for substantial efforts to reply to all the comments raised. I believe all the points raised have been answered appropriately, and I do not have any further suggestions to substantially improve the manuscript before publication. I only have three minor suggestions which the authors may wish to consider, but I do not consider them mandatory for this article to be considered for publication:

- Figure 2: you could add on the x-axis that the burst size of the specialist phage is expressed relative to that of the generalist phage, for greater clarity (same comment for Figure 6)
- Introduction (line 115 of marked-up manuscript): you could change "recapitulate" to "reproduced" or an alternative word, to avoid confusion with the more commonly used meaning of the word as "summarised"
- Discussion (line 445): linked to my major comment 3) in the first round of review, maybe rephrase "ensure that there is a cost of generalism" to "ensure that there is a cost for the generalist phage", as this would be biologically accurate, even if the cost is not due to generalism itself as you then clarify in the phrase

Reviewer #2 (Comments for the Author):

The revised version of the manuscript was substantially improved as compared to the original version. The authors' responses to my comments are satisfactory.

We greatly appreciate the additional feedback from the reviewers and the editor. We have made the suggested changes as described below. We also eliminated one supplementary table on sequencing EH7, as the complete genome annotation on GenBank is now provided.

- Please include GenBank/ENA accession numbers for the genome sequences of phages EH7 and P22vir detailed in supplementary tables 6-7.

The accession numbers for sequences from both phage have been added in the text. Please see line 591-595 in the marked-up text.

- Please also include methods details for data presented in supplementary tables 6-7.

The requested methods have been added in lines 582-590 in the marked-up text. We ultimately decided that it was most clear to remove the sequencing data from supplementary table 7, as a complete annotation is now referenced in GenBank.

- Please also ensure all references listed are included in the manuscript text. For example, I was unable to find call outs to references 49 & 50.

Thank you for noting this issue. We have been back over the references and have now moved references only discussed in the supplement into the supplementary files.

- Figure 2: you could add on the x-axis that the burst size of the specialist phage is expressed relative to that of the generalist phage, for greater clarity (same comment for Figure 6)

As suggested, we have altered the axis label to further clarify that the values represent the relative burst size of the specialist.

- Introduction (line 115 of marked-up manuscript): you could change "recapitulate" to "reproduced" or an alternative word, to avoid confusion with the more commonly used meaning of the word as "summarised"

The suggested change has been made.

- Discussion (line 445): linked to my major comment 3) in the first round of review, maybe rephrase "ensure that there is a cost of generalism" to "ensure that there is a cost for the generalist phage", as this would be biologically accurate, even if the cost is not due to generalism itself as you then clarify in the phrase

We have altered the text as suggested.

Re: mSystems01177-23R1 (Bacteriophage specificity is impacted by interactions between bacteria)

Dear Dr. William R Harcombe:

Thank you for considering my comments and those provided by reviewer. I have only a few final comments below that require attention:

1. The accession number provided for phage P22vir (RJNA1055508) does not appear to work. Please check the accession number or request public release of the data as needed.
2. Please provide the version number for breseq at line 577 of the non-marked up manuscript.
3. It appears there may still be a reference to the previous version of supplementary table 7 at line 490 of the non-marked up manuscript. Please review this.
4. I was unable to locate mention of supplementary tables 8-9, as well as the new version of table 7, in text. Please review this.

Revision Guidelines

Sincerely,

Zoe Dyson
Editor
mSystems

We are exceedingly grateful for having these issues pointed out and have addressed all of them as requested. Thank you for your careful editing.

1. The accession number provided for phage P22vir (RJNA1055508) does not appear to work. Please check the accession number or request public release of the data as needed.

This has been changed to “P22vir (reads available at SRA accession SRX22993822)”

2. Please provide the version number for breseq at line 577 of the non-marked up manuscript.

We now state “*breseq* v. 0.28 [75]”

3. It appears there may still be a reference to the previous version of supplementary table 7 at line 490 of the non-marked up manuscript. Please review this.

We have removed mention of supplementary table 7 at this location

4. I was unable to locate mention of supplementary tables 8-9, as well as the new version of table 7, in text. Please review this.

In adding citation to all supplemental tables in the main text we realized they were not cited in order, so we have changed the order. Each supplemental table is now mentioned for the first time at the line below:

Supplemental table 1 – line 205
Supplemental table 2 – line 206
Supplemental table 3 – line 217
Supplemental table 4 – line 247
Supplemental table 5. – line 262
Supplemental table 6 – line 477
Supplemental table 7 – line 483
Supplemental table 8 – line 490
Supplemental table 9 – line 504

Re: mSystems01177-23R2 (Bacteriophage specificity is impacted by interactions between bacteria)

Dear Dr. William R Harcombe:

Thank you for considering my comments. I am pleased to advise that your manuscript has been accepted, and I am forwarding it to the ASM production staff for publication. Your paper will first be checked to make sure all elements meet the technical requirements. ASM staff will contact you if anything needs to be revised before copyediting and production can begin. Otherwise, you will be notified when your proofs are ready to be viewed.

Featured Image Submissions: If you would like to submit a potential Featured Image, please email a file and a short legend to msystems@asmusa.org. Please note that we can only consider images that (i) the authors created or own and (ii) have not been previously published. By submitting, you agree that the image can be used under the same terms as the published article. Image File requirements: TIF/EPS, 7.5 inches wide by 8.25 inches tall (at least 2,250 pixels wide by 2,475 pixels tall), minimum 300 dpi resolution (600 dpi preferred), RGB, and no figure elements, e.g., arrows or panel labels. The legend should be a short description of the image, 1-2 sentences recommended.

Sincerely,
Zoe Dyson
Editor
mSystems